# Anatomy of Cerebral Arteries with Clinical Aspects in Patients with Ischemic Stroke

Francesco Barbato [1,*], Roberto Allocca [1], Giorgio Bosso [1,2] and Fabio Giuliano Numis [1,2]

1 Department of Emergency and Urgent Medicine, Stroke Unit, Santa Maria Delle Grazie Hospital, 80078 Naples, Italy
2 Department of Emergency and Urgent Medicine, Emergency Medicine, Santa Maria Delle Grazie Hospital, 80078 Naples, Italy
* Correspondence: francesco.barbato@aslnapoli2nord.it

**Abstract:** Computed tomography (CT) angiography is the main method for the initial evaluation of cerebral circulation in acute stroke. A comprehensive CT examination that includes a review of the three-dimensional and maximum-intensity projection images of the main intra and extracranial arteries allows the identification of most abnormalities and normal variants. Anatomical knowledge of the presence of any normal variants, such as fenestration, duplications, and persistent fetal arteries, plays a crucial role in the diagnosis and therapeutic management of acute stroke. However, the opposite is also true. In fact, sometimes it is the clinical picture that allows weighing how relevant or not the alteration found is. Therefore, in this review, a concise representation of the clinical picture attributable to a given arterial vessel will be included.

**Keywords:** cerebral vessels; ischemic stroke; neurovascular anatomy; cerebral arteries



## 1. Introduction

Ischemic stroke is the second most common cause of death worldwide and is a leading cause of disability, with an increasing incidence in developing countries [1]. It occurs when a blood clot blocks or narrows an artery that carries blood to the brain. The clinical management of a stroke is complex and multidisciplinary [2]. Neurological evaluation is a fundamental moment as it allows one to carry out a differential diagnosis, a prognostic, and finally, a therapeutic evaluation [2]. Therefore, knowing exactly the anatomy of the cerebral vessels is a necessary assumption to avoid gross medical errors [2,3]. Although digital subtraction angiography (ASD) remains the gold standard, computed tomography (CT) angiography is the main method for the initial assessment of cerebral circulation in acute stroke. Changes in cerebral circulation, particularly in the Willis polygon, are common [1,4]. It is important to know the appearance of these normal variants, their prevalence, and their clinical relevance. In this review paper, we will address both the anatomy of the main extra and intracranial vessels associated with acute cerebrovascular events and the clinical picture associated with them.

## 2. Description of Vascular Anatomy

We conducted a search of reviews referring to the anatomy of the cerebral vessels and their variants in the PubMed database and in the book "Anatomia Umana" by Anastasi et al. In the PubMed database, we searched through the terms [anatomy of the cerebral vessel] AND [variants of cerebral supply] AND [normal variants of cerebral vessels] with no filter. The result was 119 papers, 35 containing direct information relevant to the question. For the clinical point, a further review of the literature was carried out using PubMed and a book, Harrison's neurology in clinical medicine (2016). The literature review produced 11 papers, of which 8 were selected for this work. In both searches, case reports were excluded.

*2.1. Extracranial Circulation*

2.1.1. Arch of the Aorta

The aortic arch follows the ascending aorta and ideally begins at the level of the upper margin of the second sternocostal joint on the right. Concave downwards, its path follows an oblique plane bringing itself posteriorly and to the left on the anterior surface of the trachea, reaching the height of the body of the fourth thoracic vertebra, where it continues in the descending aorta. This level corresponds to a narrowing of the vessel, called the aortic isthmus. From the convex face of the arch of the aorta originate the right brachiocephalic trunk (TABC) or anonymous trunk, the left common carotid artery, and the left subclavian artery [5].

Clinical point: The arch of the aorta is involved in atherothrombotic disease, aneurysmal pathology, dissection, and inflammatory diseases such as aortitis. The clinical picture, often impressive, is determined by the degree and number of vessels involved. In symptomatic cases with distal embolization or aneurysm thrombosis, the clinical presentation depends on the size of embolic debris and the location of arterial occlusions. This is added to retrosternal pain, often lacerating, and neurovegetative symptomatology [6–10].

2.1.2. Supra-Aortic Vessels (SAT)

They consist of the unnamed artery, or brachiocephalic artery (BCT) on the right, common carotid artery (CCA), and subclavian artery (SA) bilaterally. These arterial vessels are most frequently arranged in seven types [11]. The most common is type 1, present in 80.9% of the population. In these cases, BCT is divided into CCA and SA on the right, while CCA and SA on the left are formed directly from the aorta. The other variants in the origin of SAT are found in 19.1% of cases. The second most common variation is type 2, the bovine arc, with an incidence of 13.6% [11,12]. Here the left CCA originates directly from the BCT. Type 3 instead has a frequency of 2.9%, in which the left vertebral artery originates directly from the aortic arch. The other 4 variants have an overall frequency of 1% [11,12]. We also include another non-exceptional variant, the formation of the right SA, not from the BCT, but directly from the arch of the aorta, completely to the left, downstream of the left SA, where it takes the name of artery "lusoria" or ARSA (aberrant right subclavian artery) (Figure 1). It passes behind the esophagus and then finds its usual path. It can be a source of dysphagia, which results from a defect in the resorption of the fourth right aortic arch during fetal life. Finally, the direct origin of the vertebral artery from the aortic arch (4%) between the CCA and the SA on the left may occur [11–13].

2.1.3. Common Carotid Artery (CCA)

It originates from the BCT on the right and the aortic arch on the left. It runs within the carotid sulcus, between the internal jugular vein that interposes laterally and the vagus nerve posteriorly, wrapped in a connective sheath called the vascular-nervous bundle of the neck. Its terminal portion is projected at the height of the body of the C4 vertebra and gives rise to the internal carotid artery (with intracranial destination) and the external carotid artery (with cervicofacial destination). The projection of the carotid ending can occur at a variable level; the most frequently occurs at the level of the upper margin of the thyroid cartilage of the larynx, between the C6 plane and the angle of the jaw [5,14].

Clinical point: The common carotid artery can be the site of atherothrombotic disease and sometimes dissection. The carotid artery most frequently affected by these processes is the left since it originates directly from the aortic arch, while the site is preferably located at the level of the bifurcations and at the origin of the collateral branches of the carotids [6,7]. Carotid dolicoarteriopathies (CDA) are even a common finding. The global prevalence of CDA is 12.9%, and carotid kinking is more frequent in females and in the left carotid axis [15]. CDA is not associated with a major occurrence of cerebrovascular events. Bilateral common carotid artery occlusions at their origin may occur in Takayasu's arteritis [6,16]. The neurological symptomatology is variable, as, in some patients, it may go unnoticed, while in other patients presents with hemiparesis and contralateral

hemianesthesia associated with dysarthria and amaurosis. Mandibular claudication may be present after chewing [6–8].

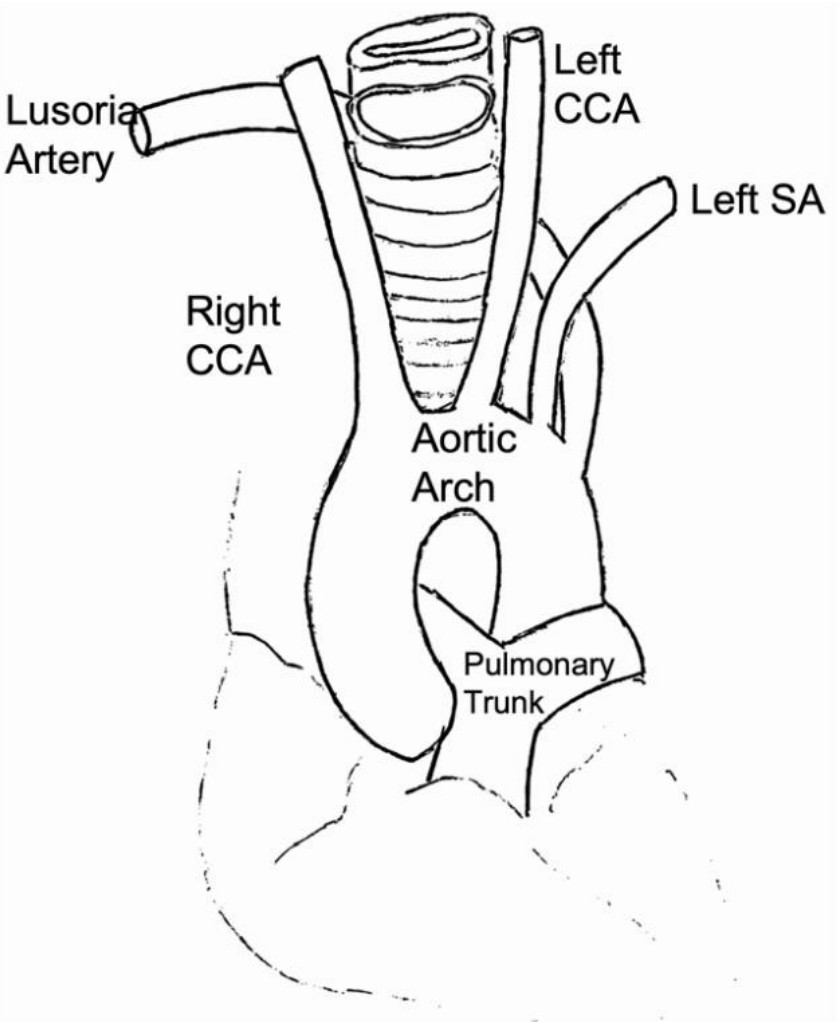

**Figure 1.** The image represents the picture typically associated with the lusoria artery. Instead of being the first branch (with the right common carotid as the brachiocephalic artery), it arises on its own as the fourth branch, distal to the left subclavian artery. It then hooks back to reach the right side with a strong relationship to the esophagus.

2.1.4. Internal Carotid Artery (ICA)—Extracranial Tract

On an anterior projection, it presents, in most cases, an outermost path of the external carotid artery at its origin. Four segments are distinguished: the cervical segment, the petrous segment, the intracavernous segment, and the supracavernous segment (Figure 2) [5]. It runs most frequently in its first segment within the maxillopharyngeal space, accompanied by the internal jugular vein. It is retropharyngeal in 10% of cases, running posterior to the wall of the oropharynx [5,17]. In any case, it then penetrates the petrous rock and runs in the carotid canal, which first has a vertical and then horizontal orientation. It returns to the intracranial plane through the torn foramen, then runs into the cavernous loggia; then, it enters the subarachnoid spaces in its supraclinoid portion. It emits on its anterior face the ophthalmic artery, then on the posterior face, spaced a few millimeters, the posterior communicating arteries, and anterior choroid. Finally, it divides to give the anterior and middle cerebral arteries [5,14]. In 20% of cases, a fetal origin of the posterior cerebral artery is recognized, which therefore originates from the internal carotid artery [18]. There are some anatomical variations. An aberrant internal carotid artery (AbICA) is a rare variation. It is considered a result of agenesis of the first cervical ICA segment. Ab-

normal vessel develops from the fusion of the inferior tympanic branch of the ascending pharyngeal artery with the caroticotympanic artery. Agenesis of ICA occurs in less than 0.01% of the population, while the bilateral absence of the artery is seen in less than 10% of cases of agenesis [13,14,19]. Prevalence of hypoplasia is 0.079% [19]. Other variants of the internal carotid artery include duplication, ICA fenestration, and high or low branching of the carotid artery (from T2 to C1 level). Rarely, ICA and ECA may originate directly from the aorta [20].

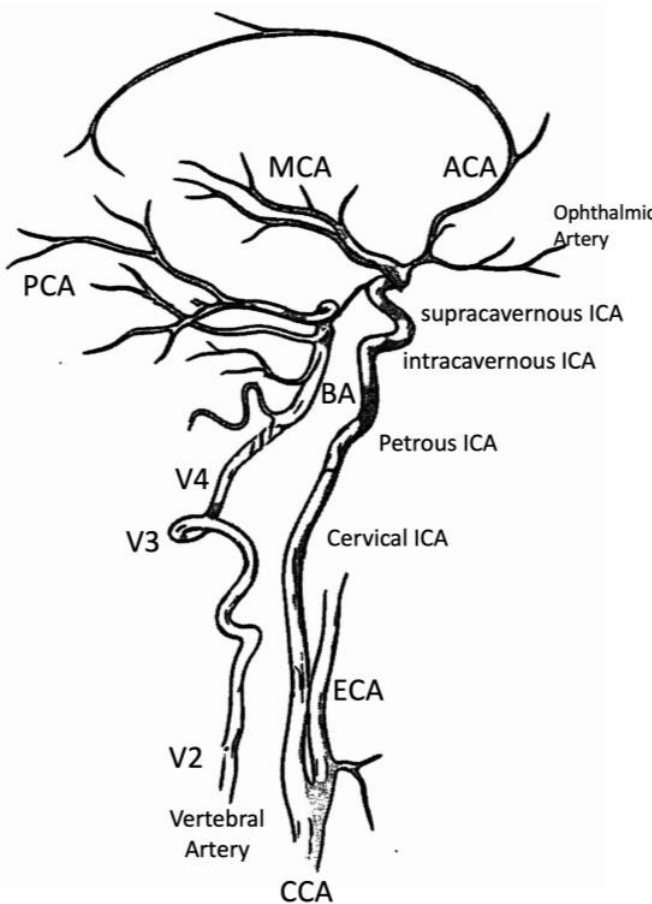

**Figure 2.** In this image are represented the main segments of the arteries afferent to the brain. On the right, it is possible to see the four segments of the ICA (cervical, petrous, intracavernous, supracavernous) with the carotid siphon and the main terminal vessels (ACA, MCA) and collateral (ophthalmic artery). On the left, we observe the vertebral artery in its intraspinal tract (V2), the loop at the level of C2 (V3), and its intracranial portion (V4) before joining with the contralateral to form the basilar artery. It is also possible to observe the origin and course of PCA.

Clinical point: The clinical picture of the occlusion of the internal carotid artery is extremely variable [6]; in fact, as in the case of a common carotid artery involvement, it could go unnoticed if the compensations determined by the Willis polygon and the carotid circles are efficient [7]. The cerebral cortex pertaining to the middle cerebral artery (MCA) is affected more frequently [6,7]. In this case, the symptoms are identical to the occlusion of proximal MCA (see the clinical point on the middle cerebral artery). When both the origins of the anterior cerebral artery (ACA) and middle (MCA) are occluded, the disorder of consciousness is accompanied by hemiplegia and complete hemianesthesia, aphasia or dysarthria, and anosognosia depending on whether the dominant hemisphere is affected or not [6]. When the posterior cerebral artery (PCA) is fetal (therefore, in 20% of cases), it can also become occluded and give rise to symptoms referable to its peripheral territory [7]. In addition to supplying the ipsilateral brain, the internal carotid artery perfuses the optic

nerve and retina through the ophthalmic artery. In 25% of symptomatic internal carotid disease, recurrent transient monocular blindness (amaurosis fugax) precedes cerebral ischemia. When retinal ischemia has been established, eye blindness can be recognized because the photomotor reflex is not present in the affected eye [6–8,16].

### 2.1.5. Vertebral Artery (VA)

The vertebral arteries are the first ascending branches of the division of the subclavian arteries and originate from the posterior and upper wall of the pre-scalenic tract. Four portions [5,21] are distinguished (Figure 2):

- V1 segment, the segment of the vertebral artery before its entry into the transverse canal (C6);
- V2 segment, part of the vertebral artery that runs in the transverse canal of C6 up to C2;
- V3 segment, which bypasses the lateral masses of C1 before redirecting medially to penetrate the atlanto-occipital membrane and through the occipital hole, makes its entry into the cranial cavity;
- V4 segment of intracranial localization, just before the meeting of the two vertebral arteries. It is noted that this V4 segment is of subarachnoid collocation.

Anatomical variants of the origin and course of the vertebral arteries are frequent. The most frequent (found in up to 2.9% of cases) is the origin of the left vertebral artery directly from the aortic arch, between the left common carotid artery and the left subclavian artery. The vertebral arteries also frequently have an asymmetrical caliber. The term "dominant vertebral artery" refers to the most voluminous (generally the left vertebral artery). It is also possible to find hypoplasia of a vertebral artery in 2–6%, characterized by a reduction in the diameter of the lumen < 2.2 mm starting from its origin [22]. Intracranial vertebral fenestrations were found in 1.1% of patients and are the most common fenestration of cerebral vessels [23,24].

Clinical point: The pathologies that can give an acute cerebrovascular syndrome starting from an extracranial vertebral artery are atherothrombosis with artery-to-artery embolism and dissection [9,10]. The formation of atherothrombotic plaques is most common in the V1 and V4 segments [10]. If the subclavian artery is occluded proximal to the origin of the vertebral artery, there is a reversal in the direction of blood flow in the ipsilateral vertebral artery. Exercise of the ipsilateral arm may increase demand on vertebral flow, producing posterior circulation TIAs, or "subclavian steal syndrome" [6,9,10].

In the case of artery-to-artery embolism, the intraarterial emboli travel to reach the ipsilateral intracranial vertebral artery and sometimes travel on to block the rostral basilar artery and/or its branches [6,10]. Dissections are mostly located in the pars transversaria segment (V2, 35% of cases) or in the atlas loop segment (V3, 34% of cases) [10,25]. Patients present with a variety of signs and symptoms, most frequently with neck pain and headache (typically occipital) as well as posterior fossa ischemic events manifesting as nausea, ataxia, dysarthria, lateral medullary syndrome, or even collapse and coma. Other presentations include spinal cord infarction and even cervical nerve root impairment [25].

### 2.2. Intracranial Circulation

### 2.2.1. Willis Polygon

The Willis polygon represents a compensatory network of the intracranial arteries. It has the shape of a heptagon whose sides are: the precommunicating segments of the anterior cerebral arteries, the anterior communicating artery, the posterior communicating arteries, and the precommunicating segments of the posterior cerebral arteries [5]. It is characterized by great variability [5,12,13]. About 22 different forms of Willis polygons have been described (Figure 3). It is complete only in 13–21% of cases [12,13].

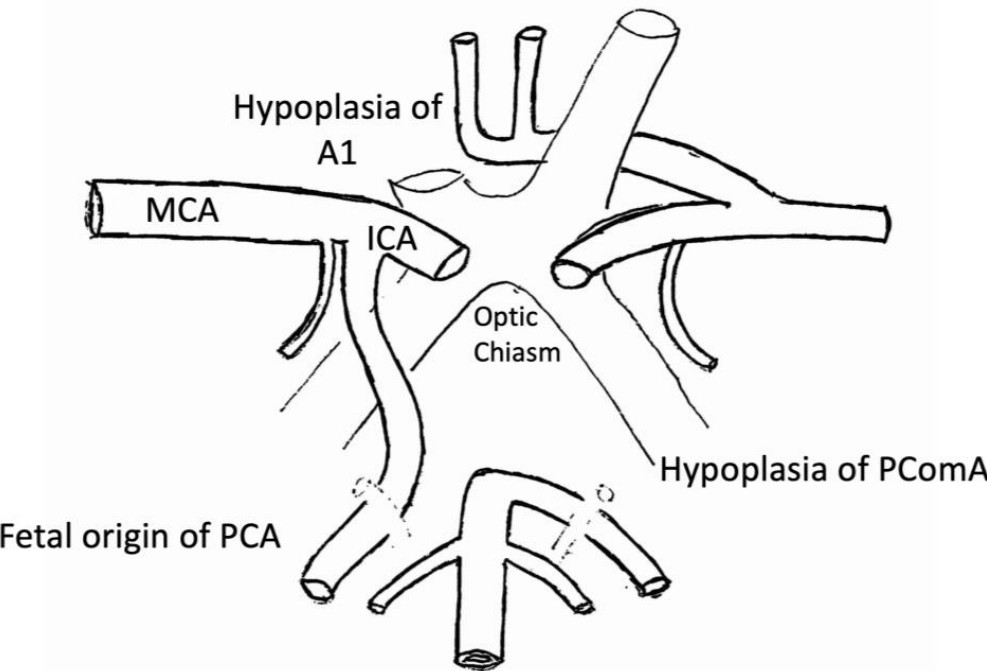

**Figure 3.** In this figure is arranged the Willis polygon (heptagon), emphasizing the possible hypoplasia/agenesis of the posterior communicating artery on the right, the fetal origin of the PCA directly of the ICA, and the hypoplasia of the left anterior cerebral artery. About 22 different types of Willis polygon are described.

Fenestration may occur in all cerebral arteries, but it most commonly involves the Willis polygon, basilar artery, and anterior cerebral artery. Fenestration is the presence of two arterial channels corresponding to a single path with partial or segmental arterial duplication. It involves the segmental division of a vessel to form two separate lumens with endothelial walls. These lumens may be surrounded by the same adventitia or have separate external laminae [26].

Among the anatomical variants, we recognize the carotid-basilar anastomoses. These anastomoses correspond to the persistence of vestigial arteries, which should normally regress during the development of the embryo, between the internal carotid artery and the vertebrobasilar system [12,13,26].

The most frequent is the persistence of the trigeminal artery (0.02–0.6%). It connects the C4 or C5 segment of the internal carotid artery with the anterior face of the middle third/upper third junction of the basilar artery. It passes through the Meckel cord after bypassing the *clivus* (parasellar course), then runs near the trigeminal nerve in its cisternal portion before anastomizing with the basilar artery. It can also pass through an orifice right into the back of the *sella turcica* (trans-sellar course). It is extremely important to recognize it as it can be crucial to plan a neurosurgical intervention aimed at the pituitary gland. Hypoplasia or absence of the homolateral posterior communicating artery and the P1 segment of the PCA is often associated with this variant. An absence/hypoplasia of the homolateral vertebral artery can also be observed, and, in this case, the basilar artery is fed by the persistent trigeminal artery. Another anastomotic persistence is the presence of the acoustic artery, which is extremely rare. It connects, crossing the petrous rock, the internal carotid artery in the carotid canal, and the caudal part of the basilar trunk. The presence of the hypoglossal artery is the second in order of frequency among carotid-basilar anastomoses (from 0.027% to 0.26%). It originates on the posterior face of the upper portion of the internal carotid artery in the cervical segment, at the height of the C1–C2 vertebral bodies. It then runs posteriorly and upwards in the hypoglossal foramen (or condylic canal), which appears enlarged. It has a rear concavity curve, then curves

to reach the midline. It is often associated with hypoplasia of the vertebral and posterior communicating arteries [27,28].

The last one we mention is the presence of the proatlantal artery. This carotid-vertebrobasilar anastomosis is rare. There are two types: the first coming from the internal carotid artery and the second from the external carotid artery. It originates from the posterior face of the internal carotid artery at the C2–C3 vertebral segments (more rarely C4) or from the proximal portion of the external carotid artery. It has a posterior and ascending course; it surrounds the lateral masses of C1 and accompanies the homolateral vertebral artery in its entry into the foramen magnum [21,27,28].

### 2.2.2. Internal Carotid Artery (ICA)—Intracranial Tract

The intracranial internal carotid artery presents, in its intracavernous portion, a series of curves that give it an Italic "S" appearance open upwards. This curved portion bears the name of the carotid siphon. Using the classification of Fischer, five segments for the carotid siphon are distinguished, named from C5 proximal to C1 distally, in the opposite direction to blood flow [29]. Segments C4 and C3 have an intracavernous localization, and segments C2 and C1 have a distribution above cavernosa (Figure 4).

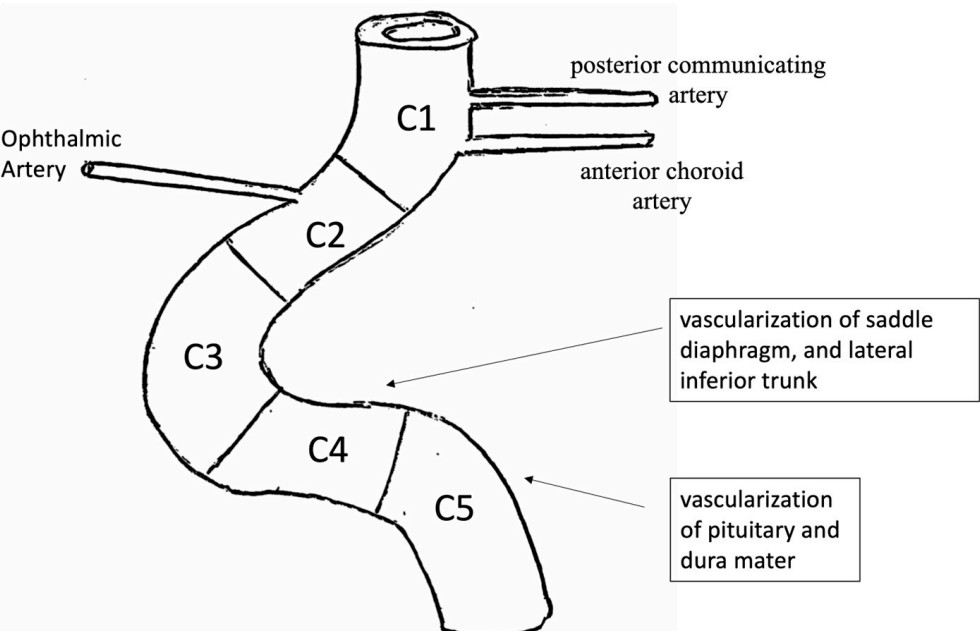

**Figure 4.** It is possible to observe Fisher's subdivision of the carotid siphon. Five segments for the carotid siphon are distinguished, named from C5 proximal to C1 distally. Segments C4 and C3 have an intracavernous localization, and segments C2 and C1 have a distribution above cavernosa.

- C5 segment gives rise to the meningo-pituitary trunk, from which emerges above all some branches that participate in the vascularization of the pituitary and adjacent dura mater;
- C4 segment has a landscape orientation and is directed forward. Its main collaterals are the capsular arteries, which vascularize the saddle diaphragm and the lateral inferior trunk. The lateral inferior trunk is divided into three branches: upper, anterior, and posterior. They participate, respectively, in the vascularization of the roof of the cavernous sinus and oculomotor nerves;
- C3 segment is curved and has a front convexity;
- C2 segment, above cavernous and infra-clinoid, is short and has a posterior, horizontal, or slightly ascending orientation. It gives rise to the superior pituitary arteries, which vascularize the anterior loggia of the pituitary gland. From the anterior part of the upper portion of the C2 segment originates the ophthalmic artery. The ophthalmic

artery is of subarachnoid localization, more rarely intradural. It has an anterior orientation to reach the optic canal, where it runs against the lateral inferior face of the optic nerve. From this segment, the intradural arteries do not possess an elastic lamina and are represented by a small tunica adventitia and poor elastic fibers in the tunica media;

- Finally, segment C1 is above cavernous and supraclinoid.

In its supra-clinoid portion (anatomically subarachnoid), the ICA is divided into four branches: posterior communicating artery, anterior choroid artery, anterior cerebral artery, and middle cerebral artery.

Clinical point: it is the same as the extracranial counterpart. The atherothrombotic disease is the most frequently encountered disease, particularly at the level of the carotid siphon. Intracranial stenosis may be present in the C1 and C2 tracts with clinical transient amaurosis and contralateral hemiparesis. At the entrance to the carotid canal, it is sometimes possible that the internal carotid artery undergoes dissection with artery-to-artery embolism. In this case, due to the low number of elastic fibers in the middle tunic and the almost absent elastic lamina in the case of dissection, it is preferable to use a double antiplate aggregation rather than an anticoagulant.

### 2.2.3. Anterior Cerebral Artery (ACA)

The anterior cerebral artery (ACA) originates from the carotid ending on its anteromedial face. It is divided, according to Fischer [30], into five segments:

- The A1 segment, known as 'pre-communicating';
- The A2 segment, which has an ascending course up to the corpus callosum;
- The A3 segment, which surrounds the knee of the corpus callosum;
- A4 and A5, which continue their course around the corpus callosum.

The ACA presents, in its proximal portion (segment A1), an anteromedial course. It bypasses the optic chiasm to reach the interhemispheric cleavage. The A1 segment gives rise to an artery that has a recurrent posterior course: Heubner's recurrent artery, which vascularizes the head of the caudate nucleus, the anteroinferior part of the inner capsule, and the anterior portion of the putamen. The recurrent artery of Heubner (RAH) is also called a median striatal artery [5,30]. The A1 segment of the left ACA is characterized by its great variability in caliber; an asymmetry is present in 55% of cases. Hypoplasia of segment A1 is present in 10% of cases [30–33]. It is defined by a diameter of less than 1 mm. It is frequently associated with aneurysms of the anterior communicating artery [32]. The ACA then bypasses the knee of the corpus callosum. It is called the "pericallosal artery". From the pericallosal artery, some cortical branches are born, among which are identified: the orbito-frontal artery, the fronto-polar artery, the medial, anterior, and posterior frontal arteries, the paracentral artery, the superior (or pre-cuneal), and inferior medial parietal arteries. It ends with the posterior peri-callous artery and gives rise in more than 80% of cases to a callose-marginal artery, which runs in the callose-marginal sulcus [5,30].

There are several anatomical variants [30–33]. The median artery of the corpus callosum, which participates in the vascularization of the corpus callosum, the bihemispheric arteries, which correspond to an ACA that vascularizes the contralateral hemisphere (Figure 5) and, finally, the origin of a single common trunk that divides at the knee of the corpus callosum, are distinguished. This is also referred to as ACA Azygos (2%). Here, the A2 segments are fused, and the anterior communicating artery is absent (Figure 5) [33].

Clinical point: Proximal ACA occlusion is usually asymptomatic due to collateral flow through the anterior communicating artery and collaterals through MCA and PCA [6,7]. The occlusion of a single A2 segment causes contralateral motor and/or sensory deficits (leg > arm and face). If both A2 segments are derived from a single anterior brain stem, the occlusion can affect both hemispheres. [7] Deep abulia (a delay in verbal and motor response) and bilateral pyramidal signs with paraparesis and urinary incontinence. Apraxia of the march is also possible [7].

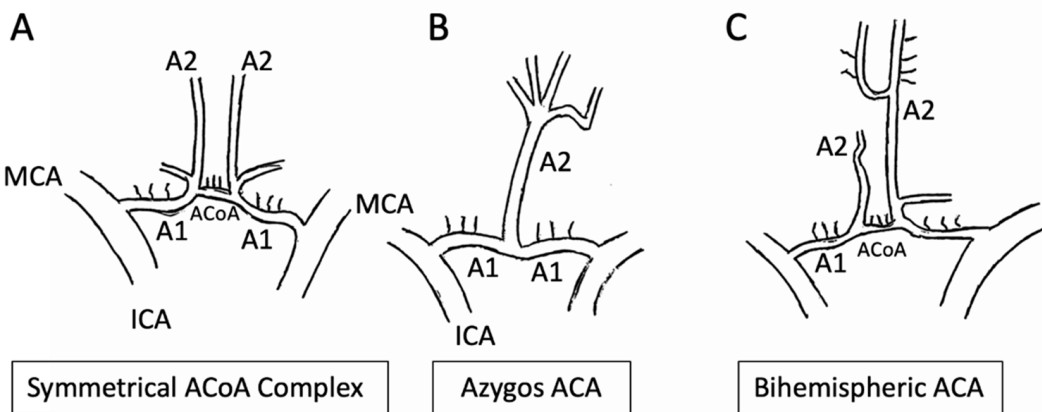

**Figure 5.** In Panel (**A**), it is possible to observe the modal arrangement of the ACA. In Panel (**B**), the origin of a single common trunk that divides at the knee of the corpus callosum, named ACA Azygos (2%), with A2 segments fused, and the anterior communicating artery is absent. The bihemispheric arteries, which correspond to an ACA that vascularizes the contralateral hemisphere, are on Panel (**C**).

### 2.2.4. Middle Cerebral Artery (MCA)

Also called the "Silvian artery", the middle cerebral artery (MCA) represents the most voluminous subdivision branch of the ICA [5]. It measures at its origin about 3 mm in diameter. Four segments are generally distinguished: the M1 segment or basal segment, the M2 segment or insular segment, the M3 segment or opercular segment, and, finally, the M4 cortical segment (Figure 6). The M1 segment extends from the origin of the artery to its entry into the Silvian cavity. It has a lateral course under the anterior perforated space. The M1 segment gives rise to the lenticolostriate branches, which arise perpendicularly and penetrate the anterior perforated space. In variable numbers (from 6 to 20), these arteries participate in the vascularization of the lenticular nucleus, the inner capsule, and the head of the caudate nucleus. They have an Italic 'S' course in frontal projection [5,19,30].

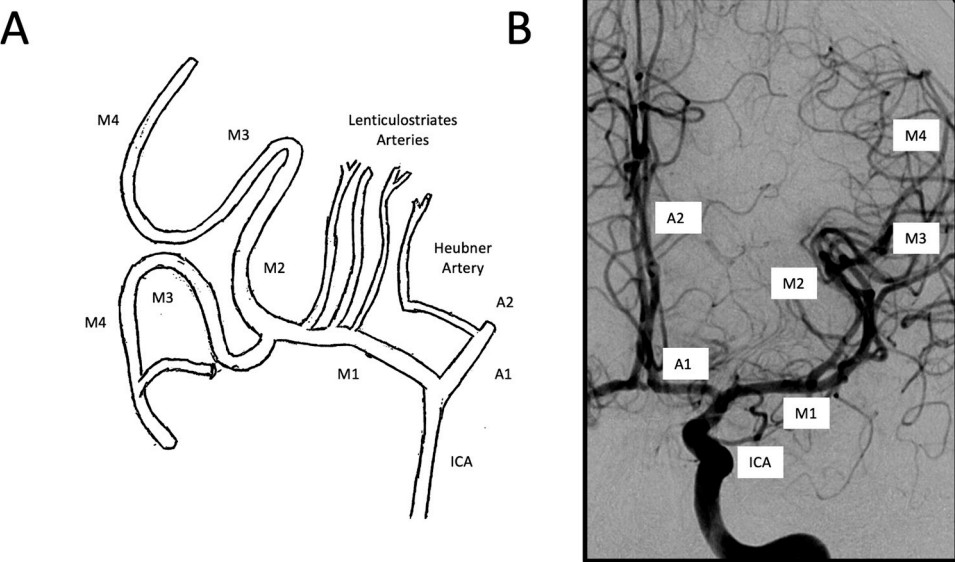

**Figure 6.** Arrangement of the middle cerebral artery in a schematization of the coronal section (panel **A**) and in the images obtained by angiography (panel **B**). It represents a terminal division branch of the ICA. Its course can be divided horizontally (MI), whose collateral branches are represented by lenticulostriate arteries. It ends in a knee where the insular segment (M2) begins. From here, two trunks are formed: upper and lower, both called M3. From the endings of both trunks of M3 cortical branches, M4 is formed.

We distinguish insular branches and cortical branches [19,30]. The insular branches, subdivision branches of the Silvian artery, curve to the lower pole of the insula to run on its outer face. They have a postero-superior oblique orientation and are divided into terminal arteries. The ascending terminal branches are distinguished, which have a first circuit with lower concavity; they are reflected on the upper line of the insula (which has a slight upper convexity), then form an upper concavity circuit before branching to the outer face of the brain. The descending branches have only an upper concavity and line the outer face of the temporal lobe. In lateral projection, the line joining the reflection points of the upper terminal branches draws a line called an "insular line", which corresponds to the upper margin of the insula and has a straight or slightly convex course upwards.

The cortical branches of the MCA are divided into three groups: ascending, posterior, and descending. From the front to the back are distinguished:

- The frontal orbital artery, which vascularizes the lower face of the frontal lobe;
- The anterior frontal artery, which vascularizes the external faces of the F2 and F3 convolutions and the frontal operculum;
- The ascending frontal artery, which runs in the central and post-central grooves. It ensures the vascularization of the ascending frontal and parietal convolutions;
- The anterior parietal artery;
- The posterior parietal artery;
- The artery of the curved fold or angular artery, which prolongs the axis of the Silvian artery. It ensures the vascularization of the posterior frontal region.

The descending branches are three:

- The anterior temporal artery, which vascularizes the temporal pole;
- The middle temporal artery, which has an oblique course at the bottom and backward in the middle part of the temporal lobe. It ensures the vascularization of the middle and posterior parts of T1eT2;
- The posterior temporal artery or temporo-occipital, which has a course roughly parallel to the previous one. It vascularizes the outer face of the occipital lobe and the back of the outer face of the temporal lobe.

We recognize as anatomical variants that the relatively rare accessory media cerebral artery (0.3–1%) corresponds to an MCA that emerges both directly from the internal carotid artery and from the homolateral ACA [34,35]. In other rare variants, MCA fenestrations typically affect the first millimeters of the M1 segment. Finally, variations in the distribution of cortical branches are very frequently found [26].

Clinical point: If the entire MCA is occluded at its origin (blocking both its penetrating and cortical branches) and the distal collaterals are limited, the clinical picture is characterized by complete contralateral hemiplegia (due to the involvement of the penetrating vessels and, therefore, of the inner capsule) (face and arm > thigh > foot), hemianesthesia, and lastly homonymous lateral hemianopsia [6,7]. When the dominant hemisphere is involved, global aphasia is also present, while when the non-dominant hemisphere is affected, anosognosia, constructive apraxia, and neglect are found. Cortical collateral blood flow and different arterial configurations are likely responsible for the development of many partial syndromes [6]. Partial syndromes due to embolic occlusion of a single branch include brachial syndrome or facial weakness with or without non-flowing Broca aphasia (frontal opercular syndrome). A combination of sensory disturbances, motor weakness, and non-fluent aphasia suggests that an embolus occluded the proximal superior division and infarcted large portions of the frontal and parietal cortices [7]. If a fluent Wernicke's aphasia occurs without weakness, the lower division of the MCA that supplies the posterior part (temporal cortex) of the dominant hemisphere is likely involved [6,8]. Sludge language and the inability to understand written and spoken language are prominent features, often accompanied by a contralateral upper quadrantanopia. Hemineglect or spatial agnosia without weakness indicates that the lower division of MCA in the non-dominant hemisphere is involved [8]. The occlusion of a lenticulostriate vessel may produce a lacunar

infarct. This produces a purely motor, purely sensory, or contralateral motor sensory picture of the injury. Ischemia inside the knee of the inner capsule mainly causes facial weakness followed by weakness of the arm and then of the legs. Alternatively, the contralateral hand may become ataxic, and the dysarthria will be prominent (clumsy hand, lacunar syndrome of dysarthria). Lacunar infarction affecting globus pallidus and putamen often has few clinical signs, but parkinsonism and hemiballism have been reported [6–8].

### 2.2.5. Anterior Choroid Artery

It originates from the posterior face of the terminal part of the ICA, a few millimeters above the posterior communicating artery. It has a fine caliber (0.6–2 mm). Its path takes place on the inner face of the temporal lobe [5]. Its course is divided into three segments: the proximal, cisternal, and distal (intraventricular) segments. After its cisternal course, where it emits branches intended for the globus pallidus, the posterior arm of the inner capsule, the tail of the caudate nucleus, the lateral ventral nucleus of the thalamus, and the red nucleus, the anterior choroid artery then follows the choroid fissure, then penetrates the temporal horn changing the orientation of its course and its caliber. It ends by branching at the level of the choroid plexuses of the lateral ventricles. It has, in its cisternal path, a posteroinferior direction, therefore, an ascending course. It has a bayonet course at the time of its entry into the ventricular horn, so it bypasses the pulvinar in its terminal portion. In frontal projection, it takes the form of an inverse "S", with the inferior medial concavity corresponding to its cisternal segment and the superior lateral concavity corresponding to its ventricular segment. The cortical territory of the anterior choroid artery is reduced to temporal branches that ensure vascularization of the anterior and medial parts of the temporal lobe [5,36].

Clinical point: Complete syndrome of anterior choroid artery occlusion consists of contralateral hemiplegia, hemianesthesia, and homonymous lateral hemianopia. However, since this territory is also provided by penetrating vessels of the proximal MCA and posterior choroid arteries, minimal deficits may occur, and patients often recover substantially [6–8]. Strokes of the anterior choroid artery are usually the result of in situ thrombosis of the vessel [7].

### 2.2.6. Anterior Communicating Artery (ACoA)

The anterior communicating artery is short, arranged transversely in front and slightly below the portion of the optic chiasm. It closes the Willis polygon [5] in front (Figure 5). This artery has numerous variations in caliber [19,24,26] (ranging up to hypoplasia that makes the two ACA independent) and in number (split in 30% of cases, tripled in 10%, exceptionally vascular network). The absence of the anterior communicating artery has been found in 5% of surgical dissections. Fenestrations in the ACoA were found in 5.3% of the population [37]. Most fenestrations were associated with 1 or more aneurysms of the AcoA [26,37].

### 2.2.7. Posterior Communicating Artery (PCoA)

It does not belong to the posterior circulation (vertebrobasilar) but to the anterior circle (carotid) due to its embryological origin [13]. It originates from the posterior face of the supracavernous ICA (C1–C2). Most often modest in size (1–2 mm on average), it measures about 10–15 mm in length. It connects the terminal portion of the ACI and the homolateral posterior cerebral artery (Figure 5). It gives rise to some thin collateral branches that participate in the vascularization of the thalamus, hypothalamus, hippocampus, optic tract, and posterior arm of the inner capsule [5,30].

### 2.2.8. Intracranial Vertebral Artery (VA)

It pierces the atlantoaxial membrane to penetrate the posterior cerebral fossa. It anastomoses with its contralateral homolog at the bulbo-pontine sulcus after bypassing the lateral face of the bulb to form the basilar artery (Figure 2). The V4 segment emits a thin

accessory branch that runs downwards and frequently anastomizes with the contralateral branch to form the anterior spinal arterial axis of the high cervical medulla [5,12,30].

Clinical point: An atherothrombotic disease of the fourth distal segment (V4) of the vertebral artery can promote the formation of a thrombus and manifest as an embolism or with the propagation of thrombosis up to the basilar artery. Proximal stenosis at the origin of PICA can cause lateral ischemia of the medulla and of the lower posterior surface of the cerebellum [9,10].

It can therefore manifest itself with a constellation of symptoms such as headache, dizziness, numbness of the ipsilateral face and contralateral limbs, diplopia, hoarseness, dysarthria, dysphagia, and ipsilateral Horner syndrome. This is called lateral (or Wallenberg's) medullary syndrome [6,9]. The occlusion of the penetrating medullary branches of the vertebral artery or PICA causes partial syndromes. Hemiparesis is not a feature of vertebral artery occlusion; however, tetra paresis can result from occlusion of the anterior spinal artery. Cerebellar infarction can present with ataxia, headache, and nausea. Distinguishing these symptoms from those of viral labyrinthitis can be a challenge, but headaches, neck stiffness, and unilateral dysmetria favor stroke [10]. It should be noted that a medial medullary syndrome rarely occurs. The cause is ischemia of the bulbar pyramid with contralateral hemiparesis of the arm and leg and sparing of the face. If the medial lemniscus and the nucleus of the hypoglossal nerve (XII c.n.) are involved, contralateral loss of the sense of joint position and weakness of the ipsilateral tongue occur [9,10]. In the intracranial dissection of the V4 segment of the spine, as opposed to its extracranial portion, there is a high risk of subarachnoid hemorrhage (up to 50% for vertebrobasilar dissections) due to the microscopic anatomy of the intracranial arteries as the elastic fibers are in the subendothelial elastic lamina, which overall is less thick [10].

### 2.2.9. Posteroinferior Cerebellar Artery (PICA)

It generally originates from the vertebral artery in its V4 portion, 15–20 mm from its end. It then runs between the tonsil and the roof of the fourth ventricle. At its peak, it forms a circuit with superior convexity. In the proximal portion of its course, it emits some bulbar branches, tonsils, and the posterior spinal arteries. In the cranial circuit of its distal portion, it gives rise to the choroid branches directed to the fourth ventricle. In its terminal portion, it generates the lower vermian branches and the hemispherical branches [5,30]. The origin of the inferior cerebellar artery is subject to many variations. It can form directly from the basilar artery (7–10%), from the internal carotid artery in its C5 portion (through a persistent trigeminal artery), or from the ascending pharyngeal artery (starting from its hypoglossal branch), but it can also arise from a common trunk that also gives rise to the middle cerebellar artery [12–14].

Clinical point: Cerebellar infarction in the PICA distribution may involve only the vermis, the lateral surface, or the entire territory of the PICA. Infarctions of the complete PICA territory are often accompanied by the formation of edema and the mass effect of increased intracranial pressure (ICP) with the headache associated with cerebellar symptoms [6]. About 15% of cerebellar infarctions of the PICA territory are accompanied by ischemia in the dorsolateral medulla (Wallenberg syndrome). The combination of lateral medullary infarction and cerebellar PICA occurs when the intracranial vertebral artery is occluded and blocks the orifice of both the PICA and the lateral medullary penetrating vessels [6,9,10]. Medial worm-limited ischemia in the medial territory of PICA usually causes a vertiginous labyrinthine syndrome that mimics peripheral vestibulopathy. Severe dizziness, ipsilateral later pulsion, and nystagmus are the main symptoms [10].

Infarcts of the PICA territory of the lateral cerebellar hemisphere are characterized by ataxia and ipsilateral lateral pulsion but without dizziness or dysarthria. Sometimes adiadochokinesia is present. When the entire cerebellar territory PICA is involved, the neurological symptomatology described above is associated with headache, usually present in the occiput on the ipsilateral side [10]. The head can also be tilted, with the occiput tending to tilt toward the ipsilateral side [6,10].

### 2.2.10. Basilar Artery

Single and median, it is formed by the conjunction of the two vertebral arteries at the height of the bulbopontine sulcus (Figure 2). It results from the fusion of the posterior longitudinal arteries. Usually of a length of 2.5–3.5 cm, it has an anterosuperior orientation. Its diameter is 3–4 mm. It runs in front of the brainstem to end at the level of the cerebral peduncles at the height of the ponto-mesencephalic sulcus, emitting the posterior cerebral arteries (PCA). It is generally rectilinear in the young subject and may have curvatures in the elderly subject. In addition to PCA, it gives rise to anteroinferior cerebellar and anterosuperior cerebellar arteries [5,12,30]. It can rarely give rise to posteroinferior cerebellar arteries. On its posterolateral face are born the circumferential branches (from four to six pairs) and the perforating branches, numerous in its distal portion. Among the anatomical variants of the basilar artery, fenestrations and hypoplasia are mentioned. Basilar artery fenestration has been found in 5% of autopsies [26,33]. Basilar artery fenestrations are mostly located in the proximal basilar trunk, close to the vertebrobasilar junction. The reported frequency of aneurysm formation in cases of basilar artery fenestration is 7%. In the latter case, the PCA shall become dependent on the carotid territory [12,26,30].

Clinical point: It is possible to distinguish the clinical picture determined in proximal or distal occlusion of the basilar artery [6,9,10,30]. *Proximal Basilar artery occlusion* most often presents as ischemia in the pons. The major burden of ischemia is in the middle of the pons, mostly in the paramedian base and often also in the paramedian tegmentum. This is because the lateral tegmentum is also supplied collateral circles coming to the PICA from the AICA and the SCA. The paramedian pontine base contains descending long motor tract and crossing cerebellar fibers. The paramedian tegmentum contains mostly oculomotor fibers. As a result, the predominant symptoms and signs in patients with basilar artery occlusive disease are motor (various combinations of hemiparesis or tetra paresis with palatal myoclonus until locked in syndrome) and oculomotor signs such as pinpoint miosis, horizontal conjugate gaze palsy or internuclear ophthalmoplegia (INO). Skew deviation of the eyes and ocular bobbing may also be present. Bulbar symptoms include facial weakness, dysphonia, dysarthria, dysphagia, and limited jaw movements. The face, pharynx, larynx, and tongue are most often involved. Alteration in the level of consciousness is an important sign in patients with basilar artery occlusion [6,10]. They may present with coma when the bilateral medial pontine tegmentum is ischemic. *Occlusion of the rostral portion of the basilar artery* (the "top of the basilar") can cause ischemia of the midbrain, thalami, and temporal and occipital lobe hemispheric territories supplied by the posterior cerebral artery branches of the basilar artery. It presents with tetraparesis, paralysis of the cranial nerves, and coma. The pupils may not be miotic, and sometimes it is possible to recognize spontaneous eye movements such as reverse ocular bobbing. This clinical picture is sometimes very difficult to distinguish from diffuse cerebral suffering of metabolic origin [10,38].

### 2.2.11. Posterior Cerebral Artery (PCA)

The posterior cerebral arteries arise from the terminal subdivision of the basilar artery at the height of the ponto-mesencephalic sulcus. The PCA has a short course in the interpeduncular cistern, so it bypasses the cerebral peduncles. About 1 cm from its origin, it receives the posterior communicating artery that anastomizes it to the internal carotid artery (Figures 2 and 3). It ensures vascularization of the internal and lower faces of the temporal lobe, the inner face of the occipital lobe and thalamus, the third ventricle, and the lateral ventricles [5,30].

The PCA divides to give rise to the internal occipital artery, ascending and the inferior temporal descending branch. The PCA is divided into four segments:

- The P1 or precommunicating segment;
- The P2 segment, which bypasses the cerebral peduncle through III c.n;
- The P3 segment, which runs along the underside of the temporal lobe;
- The P4 segment that arrive at the calcarine fissure.

The most frequent anatomical variant is the persistence of fetal organization (20%). In this case, the PCA arises directly from the back of the ICA, and the P1 segment has agenesis or hypoplasia [18]. This form corresponds to a failure to regression of modal vascularization in the fetus. In fact, during fetal life, the PCA is fed by ICA through the segment that corresponds to the future posterior communicating artery. This segment regresses in adulthood, and the P1 portion then takes charge of the vascularization of the PCA [18]. In any case, *diencephalic branches* are present. These correspond to thin branches that originate from the PCA or from the posterior communicating artery [5,12,30]. They are mainly distinguished [5,30]:

- The inferior dorsomedial arteries, which vascularize the lower portion of the thalamus and originate from the posterior communicating artery;
- The posteromedian choroid artery, which originates from the P2 segment of the ACP. It runs along the upper margin of the posterior communicating artery, then on the upper edge of the thalamus, emitting branches that vascularize the anterior nuclei of the thalamus. Finally, it penetrates the choroid canvas of the third ventricle, which it sprays;
- The posterolateral choroid artery. Ensures the vascularization of the choroid plexuses. It crosses the inner face of the pulvinar and penetrates the choroid fissure of the lateral ventricle;
- The posterior perichallose artery, which runs on the posterior face of the splenium of the corpus callosum.

Clinical point: PCA syndromes usually result from atheroma formation or emboli that lodge at the top of the basilar artery; posterior circulation disease may also be caused by dissection of either the vertebral artery or fibromuscular dysplasia. Two clinical syndromes are commonly observed with occlusion of the PCA:

- P1 syndrome: midbrain, subthalamic, and thalamic signs, which are due to disease of the proximal P1 segment of the PCA or its penetrating branches (thalamogeniculate, Percheron, and posterior choroidal arteries);
- P2 syndrome: cortical temporal and occipital lobe signs due to occlusion of the segment distal to the junction of the PCA with the posterior communicating artery.

In P1 syndromes, infarction usually occurs in the ipsilateral subthalamus and medial thalamus and in the ipsilateral cerebral peduncle and midbrain. A third nerve palsy with contralateral ataxia (Claude's syndrome) or with contralateral hemiplegia (Weber's syndrome) may result [6,9]. The ataxia indicates involvement of the red nucleus or dentatorubrothalamic tract; the hemiplegia is localized to the cerebral peduncle. If the subthalamic nucleus is involved, contralateral hemiballismus may occur [6,9]. Occlusion of the artery of Percheron produces paresis of upward gaze and drowsiness, and often abulia [39]. Extensive infarction in the midbrain and subthalamus occurring with bilateral proximal PCA occlusion presents as coma, unreactive pupils, bilateral pyramidal signs, and decerebrate rigidity [6,9]. Occlusion of the penetrating branches of thalamic and thalamogeniculate arteries produces less extensive thalamic and thalamocapsular lacunar syndromes [39]. The thalamic Déjerine-Roussy syndrome consists of contralateral hemisensory loss followed later by an agonizing, searing, or burning pain in the affected areas. It is persistent and responds poorly to analgesics [9].

In P2 syndromes, occlusion of the distal PCA causes infarction of the medial temporal and occipital lobes occurs. Contralateral homonymous hemianopia with macula sparing is the usual manifestation. Occasionally, only the upper quadrant of the visual field is involved. If the visual association areas are spared, and only the calcarine cortex is involved, the patient may be aware of visual defects. Medial temporal lobe and hippocampal involvement may cause an acute disturbance in memory, particularly if it occurs in the dominant hemisphere. The defect usually clears because memory has a bilateral representation. If the dominant hemisphere is affected and the infarct extends to involve the splenium of the corpus callosum, the patient may demonstrate alexia without agraphia. Visual agnosia for

faces, objects, mathematical symbols, and colors and anomia with paraphasic errors (amnestic aphasia) may also occur in this setting, even without callosal involvement. Occlusion of the posterior cerebral artery can produce peduncular hallucinosis (visual hallucinations of brightly colored scenes and objects) [6–10,30].

Bilateral infarction in the distal PCAs produces cortical blindness (blindness with preserved pupillary light reaction). The patient is often unaware of the blindness or may even deny it (Anton's syndrome). Bilateral visual association area lesions may result in Balint's syndrome, a disorder of the orderly visual scanning of the environment, usually resulting from infarctions secondary to low flow in the "watershed" between the distal PCA and MCA territories, as occurs after cardiac arrest. Patients may experience the persistence of a visual image for several minutes despite gazing at another scene (palinopsia) or an inability to synthesize the whole of an image (simultagnosia) [6,30].

### 2.2.12. Anteroinferior Cerebellar Artery (or Middle Cerebellar Artery, AICA)

It originates in the lower third of the basilar artery in more than 3/4 of cases and in the vertebrobasilar junction only in about 10% of cases [40]. From its origin, the artery runs laterally and inferiorly on the front face of the bridge, giving rise to multiple perforating branches that vascularize the lower 2/3 of the bridge and the upper part of the medulla oblongata. The artery then enters the cistern at the angle of the cerebellum and near the facial nerve and vestibulocochlear nerve. It then heads laterally, in a higher position than the cerebellar flocculus, then folds medially, crossing the lower face of the cerebellar hemispheres, where it sends some collateral branches that vascularize these structures, as well as portions of the lower part of the worm and, in depth, the homolateral toothed nucleus [5,40].

Clinical point: The symptomatology is like that of lateral medullary syndrome but with peripheral paralysis of the facial nerve (VII c.n) and of the eighth cranial nerve (impairment of homolateral hearing to the lesion) [6,10].

### 2.2.13. Anterosuperior Cerebellar Artery (or Superior Cerebellar Artery, SCA)

It originates from the terminal portion of the basilar artery. Immediately it moves laterally, just below the oculomotor nerve, which separates it from the posterior cerebral artery, and turns around the brain stem, near the trochlear nerve, to reach the cerebellomesencephalic fissure reaching the upper surface of the cerebellum. Here it is divided into branches into the pia mater that anastomize with those of the anterior inferior cerebellar artery (AICA) and posterior (PICA). In a percentage of subjects between 12% and 14%, the artery, once originated, can go to duplicate [5,13,40].

Clinical point: The clinic is characterized by dynamic ataxia, dysmetria, dysarthria, dizziness, and headache [6,10].

## 3. Discussion

The WHO (World Health Organization) defines stroke as "a sudden appearance of signs and/or symptoms referable to deficits in brain functions, localized or global lasting more than 24 h or to an unfortunate outcome not attributable to any other apparent cause than cerebral vasculopathy" [41].

As a prerequisite for ischemic stroke, therefore, there is the occlusion of a cerebral artery, resulting in a reduction in blood supply and, therefore, oxygen and nutrients to the brain tissue. This causes alterations in the functionality of brain cells that quickly lead to necrosis of the portion of brain tissue most affected by ischemia (the ischemic core), around which there is a portion of tissue in ischemic suffering but still recoverable (the ischemic penumbra). The mechanisms that induce cell death (homeostasis of $Ca^{2+}$, $Na^+/Cl^-$, $K^+$, excitotoxicity, peri-infarction depolarization, oxidative stress, inflammation, etc.) are not limited to neurons. They involve the whole of the neurovascular unit, oligodendrocytes, and microglia [30,42].

In addition, systemic immunodepression caused by ischemic stroke is responsible for infections that worsen the functional prognosis. [43]. Therefore, it becomes essential to recognize and distinguish neurological symptoms attributable to the involvement of a brain area dependent on a vessel. This is not so much for a mere anecdotal purpose but for its clinical and prognostic implication since, in this way, it is possible to choose the best possible therapeutic option. Historically, studies of the clinical consequences of strokes, and their relation to vascular territories in the brain, provided information about the location of various brain functions [43]. Now, non-invasive functional imaging techniques have largely supplanted the correlation of clinical signs and symptoms with the location of tissue damage observed at autopsy [42,43]. This also helps in understanding the clinical significance of possible anatomic variants found in patients.

## 4. Conclusions

The anatomy of the cerebral circulation and the main extracranial vessels involved in cerebral ischemia is subject to significant variability. Knowledge of normal anatomy and its variants is the essential premise in the clinical approach to the stroke patient. In fact, the neurological clinical picture is strongly dependent on the vessels affected by the pathological process.

The methodical use of all patients suspected of having an ischemic stroke, of radiological investigations such as CT angiography, perfusional CT, and sometimes even cerebral angiography will allow the deepening of the anatomy of the vessels of the cerebral district and, at the same time, relating it with the clinical picture.

## 5. Future Directions

The preliminary anatomical knowledge, the mode of representation, and the incidence of vascular variations of the brain help us as a future perspective in understanding the vascular model. If, on the one side, thanks to the study of the neurovascular unit, we understand more and more the pathophysiological mechanisms underlying ischemic stroke, with the aim of reaching new therapeutic targets. On the other hand, emergency imaging methods allow us to expand our knowledge of the macroscopic anatomy of the cerebral vessels and to relate it to the clinical picture of a patient in urgency. In addition, anatomical variations such as fenestrations, agenesis, hypoplasia, or the aberrant nature of vessels provide the essential prerequisite for surgeons before planning neurovascular surgeries.

**Author Contributions:** Conceptualization: F.B. Methodology: F.B. and G.B. Software: R.A. Validation: F.G.N. Formal analysis: F.G.N. Investigation: F.B. and R.A. Resources: F.G.N. Data curation: G.B. Writing—original draft preparation: F.B. Writing—review and editing: F.B. Visualization: All authors. Supervision: F.G.N. Project administration: F.G.N. All authors have read and agreed to the published version of the manuscript.

**Funding:** This research received no external funding.

**Institutional Review Board Statement:** Not applicable.

**Informed Consent Statement:** Not applicable.

**Conflicts of Interest:** The authors declare no conflict of interest.

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
