# Peer review of "Anatomy of Cerebral Arteries with Clinical Aspects in Patients with Ischemic Stroke"

_2813-0545, doi:10.3390/anatomia1020016_

Round 1

Reviewer 1 Report

This article is well written and provides a thorough explanation for all of the anatomical variations cited. There are several times when acronyms are explained but then not used throughout the articles. I have attempted to find all of the areas where they need to be put into the article, but please make sure to read through and change all that are needed. There are minor edit suggestions throughout the edited article I have provided but the key thing that also needs to be updated is the following:

The plagiarism noted in my recommendation to the editor is a quote found on line 629. There is a quote used but it is not cited so this must be cited or it can be viewed as plagiarism.

Author Response

thank you very much for your analysis. I proceeded to make the changes he suggested

Reviewer 2 Report

Barbato et al. have submitted a paper on the investigation of cerebral arteries in stroke. The paper has an anatomical and semiological approach and the topic has potential. However, the paper is significantly flawed due to the methodology and general style, specifically:

- the authors have presented the "materials and methods" section as if they performed a systematic review; however, the databases are insufficient (no other major databases or gray literature) and the search terms are inadequate. no mesh terms are used, the phrasing is incomplete and inaccurate, and the use of operators is wrong. this is the reason why the authors had only 27 results where one would expect to find hundreds of articles.

- the fact that the authors decided to limit the discussions to be based on 6 papers and a book is unacceptable

- to be clear, this is not a systematic review, therefore the structure of the paper should be that of a review paper, with sections titled accordingly

- most of the anatomical information on the various vessels is far too basic to be included, as it is general data known to all/the majority of the readers of the paper; the schematics are fine, and most of the details (origin, trajectory, reports) are already visible in the figures and are redundant in the text; the discussions on the anatomic variants are acceptable, but should be adequately referenced

- figures with multiple fragments should be properly presented in the caption, as per Instructions per Authors

- the manuscript should be significantly shortened, perhaps to half its present size (mainly by restricting the anatomical description of the vessels, as prior mentioned)

- there are some grammar and spelling errors throughout the manuscript; please revise thoroughly

Author Response

I thank his advice and recommendations. Although tough they were very productive and I really thank you very much for your analysis. I corrected the search terms, I revised the section setting it as a simple review, I modified the images dividing them in case it was more than one present at the same time, and finally I tried to lighten the whole article a little while always maintaining my purpose of be accessible to all (not just to readers of this magazine who are experts in these fields)

Reviewer 3 Report

This article provides a lot of useful information on anatomical variants of the aortic and cerebrovascular system. There are several issues that must be addressed, mostly because the relevance of stroke in this paper is not strong and many important variants are still missing. 

The title should be rephrased and the term “semeiology” should be avoided. I encourage the authors to remove anything related to “stroke” from the title too. Because the term “stroke” is only mentioned twice in the results. So it is quite easy to tell that the main emphasis of this paper should not centralize around stroke. I kindly suggest the authors to rather focus on all vascular variants and try to be as comprehensive as possible.

There are seven types of aortic arch variants, and their prevalence was accurately reported in this recent meta-analysis: https://doi.org/10.1016/j.jvs.2017.06.097. The authors should also be aware of the thyroid ima artery, a very common variant which often originates from the aortic arch (https://doi.org/10.1016/j.aanat.2021.151803)

When reporting the prevalence of any variations, relevant papers must be cited. I notice that there are numerous sentences that lack citation including, but not limited to, sentences in lines 71-72, 73-74, 97-98, 116-117, 123-124, and etc. I stop checking because it is throughout the whole paper. This issue really limits the credibility of the paper! Try to cite meta-analyses whenever possible because the prevalence is more accurately studied. 

Other variations in which the authors might be interested include:

- Transverse anastomosis of the vertebral arteries (https://pubmed.ncbi.nlm.nih.gov/24068398/ and https://www.ncbi.nlm.nih.gov/pmc/articles/PMC3962325/)

- Persistent stapedial artery (http://www.ajnr.org/content/early/2020/09/03/ajnr.A6738)

- Duplication of anterior communicating artery is a well-known variant

- Accessory middle cerebral artery (http://www.ajnr.org/content/10/3/563.short)

- Fenestration of anterior communicating artery

Author Response

thank you very much for your analysis and for the very useful advice. I proceeded to modify the article following his advice, including citation (now inside the article) and the title as it did not include "all strokes". According to its modification, it now refers to clinical aspects secondary to the stop of flow of arterial vessels (ischemic stroke), and this seems to me to be really congruous. Thank you

Reviewer 4 Report

The authors provide a review of essential cerebral brain anatomy associated with the corresponding clinical semiology in case of infarction. The paper is well organized and interesting to all clinicians who deal with acute stroke patients. I have only some minor comments. 

I suggest providing references not on the title of each chapter but in important parts of the text , particularly when presenting epidemiological data, percentage of variants etc. For example, provide references for:

·      Agenesis of ICA occurs in less than 0.01 (r 128)

·      Prevalence of hypoplasia is 0.079% (r 130)

·      25% of symptomatic internal carotid disease, recurrent transient monocular 146 blindness (amaurosis fugax) precedes cerebral ischemia. (r 146)

·      2-6%, characterized by a reduction in the diameter of the lumen < 2.2 mm starting from its origin  (raw 169)

·      

·       

ICA may originate directly from the aorta”. In that cases ECA is absent or ECA/branches originate from the homolateral vertebral artery?

Raw 137 : “If the thrombus propagates along the internal carotid artery in the MCA  or the embolism …”. Please rephrase.

R265. I suggest adding that ophthalmic artery is an essential anatomic boundary regarding dissections. After C2 segment where ophthalmic artery rises, intradural arteries have no external elastic lamina, little adventitia and a paucity of elastic fibers in the media , which make the arteries more prone to rupture and provoke a subarachnoid hemorrhage in case of a dissection. 

This boundary is used by clinicians in order to differentiate between intracranial and extracranial dissections. Finally, this certain anatomical hint has also clinical implications regarding antithrombotic treatment, for example anticoagulants or intravenous thrombolysis have relative contraindication in intracranial dissections due to the risk of SAH-while not at extracranial dissections. (see Lancet Neurol 2015; 14: 640–54 )

Please make the appropriate corrections: 

Raw 288 "perichallose artery"  ïƒ¨ pericallosal artery?

Raw 373 non-flowing ïƒ¨ non-fluent aphasia

Raw 380 emineglect ïƒ¨ hemineglect

Raw 457 it realizes a caudal circuit around the cerebellar tonsil . Please rephrase.

480 later pulsion 

505 tegmen ïƒ¨ tegmentum

Author Response

thank you very much for your comments. I found them really interesting and useful, even from a clinical point of view it was important to underline the difference of a C2 level dissection. therefore I proceeded to modify the article according to his advice. a thousand thanks

Round 2

Reviewer 2 Report

The authors have performed some corrections to the paper. There are still some issues:

- Section 2 is named "Relevant Section" which is not very useful - the authors should rename it to a clearer title, such as "Description of vascular anatomy" or the sorts.

- The subsections of chapter 2 should be formatted according to the instructions for authors (with subheadings) and the numbers in brackets (reference numbers, I presume) should be removed

- There are still some language errors in the manuscript (" Utilizzando la classificazione di", "In questo caso per lo scarso...") - please correct the errors throughout the paper, using an english editing service, if needed, or a native speaker

- the section Conflicts of Interest in the back matter should be adequately filled in

Author Response

thank you very much for your advice, very useful. I proceeded to make the necessary changes according to the instructions. Thank you

Reviewer 3 Report

The manuscript has much been improved and is now acceptable for publication. Congratulations to the whole team!

Author Response

Thank you !